# Moonlight Is Perceived as a Signal Promoting Genome Reorganization, Changes in Protein and Metabolite Profiles and Plant Growth

**DOI:** 10.3390/plants12051121

**Published:** 2023-03-02

**Authors:** Jeevan R. Singiri, Govindegowda Priyanka, Vikas S. Trishla, Zachor Adler-Agmon, Gideon Grafi

**Affiliations:** French Associates Institute for Agriculture and Biotechnology of Drylands, Jacob Blaustein Institutes for Desert Research, Ben-Gurion University of the Negev, Beersheba 8499000, Israel

**Keywords:** full moonlight, genome organization, proteome, metabolome, photoreceptors, lunar farming, post-germination growth, *Brassica juncea*, *Nicotiana tabacum*

## Abstract

Rhythmic exposure to moonlight has been shown to affect animal behavior, but its effects on plants, often observed in lunar agriculture, have been doubted and often regarded as myth. Consequently, lunar farming practices are not well scientifically supported, and the influence of this conspicuous environmental factor, the moon, on plant cell biology has hardly been investigated. We studied the effect of full moonlight (FML) on plant cell biology and examined changes in genome organization, protein and primary metabolite profiles in tobacco and mustard plants and the effect of FML on the post-germination growth of mustard seedlings. Exposure to FML was accompanied by a significant increase in nuclear size, changes in DNA methylation and cleavage of the histone H3 C-terminal region. Primary metabolites associated with stress were significantly increased along with the expression of stress-associated proteins and the photoreceptors phytochrome B and phototropin 2; new moon experiments disproved the light pollution effect. Exposure of mustard seedlings to FML enhanced growth. Thus, our data show that despite the low-intensity light emitted by the moon, it is an important environmental factor perceived by plants as a signal, leading to alteration in cellular activities and enhancement of plant growth.

## 1. Introduction

The moon is the only natural satellite of our planet, orbiting elliptically around Earth in about 29.5 days. This cycle is known as the lunar cycle, and can be broken into four major phases, namely, new moon, first quarter, full moon, and last quarter. The moon reflects sunlight at a very low intensity, which is negligible even at its peak and far below the level of photosynthetically active radiation (PAR) required to support photosynthetic growth of organisms on the ocean or land surface [1]. Although not sufficient for photosynthesis, the moonlight and the lunar cycle can affect the behavior of vertebrate and invertebrate species, including reproduction, communication, foraging and predation [2]. Despite its low intensity, full moonlight can be absorbed even by symbiotic corals (e.g., cnidarian/zooxanthellate symbioses) via photoreceptors in the cnidarian hosts that allow them to sense and respond to low levels of blue moonlight [3]. It is believed that rhythmic exposure to moonlight affects the life cycle of plants, from seed germination to fruit maturation and dispersal [4]. A fundamental tenet in traditional lunar farming states that “above ground crops” should be planted on the days between the new moon and the full moon while “below ground crops” should be planted between the full moon and the next new moon, though these lunar farming practices have been criticized for lacking strong scientific support [5]. Notably, every month, plants may be exposed to 80–100% of the full moonlight for more than 5 h each night (Appendix A) during 8 consecutive nights, and surprisingly, studies on the impact of this prominent environmental factor on plant biology and ecology are relatively meager. Some scientific studies described a correlation between the lunar cycle and plant growth and development and the state of the moon at the time of sowing, which appeared to play an important role in seed germination, vegetative growth and flowering [6]. For example, early studies showed that the seeds of multiple crop plants, including vegetables and cereals, sown 2 days before the full moon displayed better germination and post-germination growth and produced a better harvest than those emerging from seeds sown 2 days before the new moon [7]. Multiple mechanisms have been suggested to explain the effect of moonlight on plant growth, including the breakdown of starch by diastases [8], variations in water absorption during seed imbibition [9,10] and gaseous exchange [11], as well as the effect of the moon on the movement of sap in plants [12]. Some recent scientific reports highlighted a possible link between the lunar cycle and plant physiology and phenology. Accordingly, a study of *Cereus peruvianus* (Peruvian apple cactus) showed that under a long-day photoperiod, large-sized flowers open almost exclusively at night, in a 24 h rhythm, over a course of 3–4 days that span the cycle of the full moon [13]. Strong evidence for the effect of moonlight on plant biochemistry and molecular biology came from a recent study revealing a massive transcriptional variation in *Coffea arabica* under full moonlight (FML) conditions. Among the genes affected by FML are core clock genes, stress-responsive genes and the photoreceptor phototropin1 (*PHOT1*). The enhanced expression of multiple stress-responsive genes, such as redox genes and heat shock protein genes (*HSPs*), suggested that FML is perceived by the plant as a stress signal [14].

Conceivably, the sunlight emitted by the moon at an essentially similar spectrum might be acting as an environmental signal, rather than an energy source, which is perceived by the plant, most likely via photoreceptors, to induce variation in cellular function. To gain better insight into how plants interpret the moonlight signal, we examined the response of tobacco plants to FML. Therefore, tobacco seedlings exposed to FML were analyzed for nuclear morphology, the methylation status of repeats and their proteomic and metabolomic profiles compared to dark-treated plants. In addition, we examined the effect of FML on the post-germination growth of the crop *Brassica juncea* (Indian mustard). Significant changes were found in all the parameters studied, highlighting the potential role of moonlight in controlling plant growth and development.

## 2. Results

### 2.1. Exposure to FML Induces Epigenetic Variation

Tobacco seedlings were subjected to darkness or to FML for 1 h or 5 h. Leaves were collected and fixed immediately in acetic acid–ethanol. Light-grown tobacco seedlings collected 5 h after the dark period were used as a reference. Nuclei prepared from fixed leaves were stained with DAPI and visualized under a confocal microscope. The results show a significant increase in nuclear size (diameter) following exposure to FML for 5 h in comparison to darkness that was comparable to the nuclei size of light-grown plants (Figure 1A,B). The size of nuclei prepared from 1 h FML-treated plants was indistinguishable from that of dark-treated plants. Accordingly, the average nuclear diameter for plants treated with 1 h of dark, 5 h of dark, 1 h of FML, 5 h of FML and light was 14.9, 15, 15.6, 20.9 and 21.4 μm, respectively. The ~1.39-fold increase in nuclear diameter under 5 h of FML and light compared to dark treatment accounted for a ~2.37-fold increase in nuclear volume (considering the nucleus as a sphere, V = 4/3πr^3^). Our data are consistent with a previous report demonstrating changes in the nuclear architecture of *Arabidopsis* upon the transition from dark to light, which might reflect genome reorganization accompanied by chromatin decondensation [15]. A similar increase in nuclear size was obtained with *B. juncea* seedlings exposed (10 September 2022) for 5 h to the dark, FML and growth room light (Appendix A). The results show a ~1.26-fold increase in nuclear diameter under FML and light, which accounted for a ~2-fold increase in nuclear volume compared to dark-treated plants.

To assess the involvement of DNA methylation in genome restructuring, we analyzed the state of methylation of the tobacco retrotransposon Tto1 and repetitive DNA sequences, namely the sub-telomeric HRS60 repeats [16] and the centromeric repetitive sequence RS-3-19 [17]. Thus, genomic DNAs prepared from the leaves of tobacco plants subjected to FML and darkness for 5 h, or from plants grown in the light for 5 h following the dark period, were digested with methylation-sensitive enzymes *Hpa*II and *Msp*I followed by PCR amplification using primers for Tto1, HRS60 and RS-3-19. The results show (Figure 1C) no change in the methylation status of the Tto1 and HRS60 in all treatments examined. However, a significant reduction in CG methylation at centromeric RS-3-19 repeats was observed under FML and light, but not under dark treatment, since very low level of the RS-3-19 fragment could be obtained from *Hpa*II digest.

The dynamic modification of histone H3 was monitored using antibodies to dimethylated lysine 4 (H3K4me2) and K9 (H3K9me2). The results show (Figure 1D,E) that during the light period, two H3 isoforms di-methylated at K4 and K9 were evident, namely, a major isoform at the position corresponding to ~15 kDa and a fast-migrating isoform (H3FMI) at a position of about 12 kDa. Interestingly, the H3FMI was not detected during the dark period, but re-established upon exposure to FML for 5 h. We assumed that the H3FMI resulted from a cleavage at the C-terminal region of H3. Indeed, Cell Signaling antibodies raised against the C terminus of histone H3 (αH3-Cter, Figure 1F) detected the 15 kDa H3 isoform but failed to detect the H3FMI, indicating that this isoform is lacking a portion of the C-terminal region.

### 2.2. Exposure of Tobacco Plants to FML Induces Changes in Metabolic Profile

We compared the metabolic profiles of plants exposed to FML for 1 and 5 h with plants subjected to darkness for 1 and 5 h using GC-MS and identified 98 primary metabolites (Appendix A); 75 differentially expressed metabolites (DEMs) were identified in all the pairwise comparisons between treatments (FDR-adjusted *p* value < 0.05). A principal component analysis (PCA) of the 75 DEMs separated them into four groups according to treatments, showing that exposure to FML had a significant effect on the metabolites accumulated in leaves, which was also dependent on the exposure time (Figure 2A). A heatmap of differentially expressed metabolites demonstrates clear differences in the expression of certain metabolites following 1 and 5 h of exposure to FML compared to darkness (Figure 2B). Interestingly, while the level of multiple amino acids was reduced following 1 h of exposure to FML, their levels were significantly elevated after 5 h of exposure to FML, except for the amino acid lysine, whose level was significantly reduced (Table 1). The level of the sugar raffinose increased significantly after 5 h but not after 1 h of exposure to FML.

We were aware that potential light pollution (LP) might introduce experimental errors, and thus, we run experiments with tobacco plants at the new moon stage (NM, dark nights without moonlight) to examine the potential effect of LP on the nuclear morphology and metabolic profiles in comparison to dark-treated plants. The results show (Appendix A) that the metabolic profiles (Appendix A) of dark- and NM-treated plants were indistinguishable (Appendix A), and no differences in nuclear size between NM and dark treatments could be observed (Appendix A), thus excluding the possibility of potential LP at the experimental site contributing to plant response.

### 2.3. Exposure to FML Induces Changes in Proteome Profile: Upregulation of Photoreceptors and Stress Proteins

Since a long exposure to FML (5 h) resulted in notable variations in epigenetics and metabolic profile, proteome analysis was performed on proteins extracted from tobacco leaves treated with 5 h of darkness and 5 h of FML. This analysis revealed 3737 expressed proteins (Appendix A), and after filtering out potential contaminants, reverses, and those only identified by site as well as filtering for proteins with a minimum of two peptides and at least 20% coverage, we documented 2019 expressed proteins (Appendix A). A principal component analysis (PCA) of the 2019 proteins separated the dark treatment (dark—5 h) from FML (FML—5 h) with PC 1, explaining 45.5% of the variance (Figure 3A). After applying imputation, we found that among the 2019 proteins, 31 proteins were differentially expressed (DEP; FC > 1.5; *p* < 0.05, Appendix A), that is, 22 and 9 proteins were upregulated and downregulated, respectively, under FML (Appendix A). A partial list of DEPs (FC > 2) is given in Table 2. Protein class categorization of DEPs showed that the upregulated proteins include metabolite-modifying enzymes, chaperons and proteins involved in translational regulation (Figure 3B). Biological process categorization highlighted proteins involved in developmental and metabolic processes and in the response to stress (Figure 3C). FML-induced proteins include the photoreceptors Phytochrome B (PhyB) (30.6 FC) and Phototropin-2 (Phot2) protein (3.5 FC). In addition, multiple stress-responsive proteins, such as chaperons (heat shock proteins, HSP70), chaperonins and reactive oxygen species (ROS)-detoxifying enzymes (peroxidases and superoxide dismutase), were also upregulated under FML (Table 2).

The increased protein levels of PhyB and Phot2 observed under FML compared to darkness could have resulted from FML induction of *PHYB* and *PHOT2* gene expression. Thus, we first analyzed the mRNA level of *PHYB* and *PHOT2* via semiquantitative PCR (Figure 3D), followed by real-time PCR (Figure 3E) using cDNA derived from total RNA extracted from light-, dark- and FML-treated plants. The results show increased levels of *PHYB* and *PHOT2* mRNAs under FML compared to the dark. Light-grown plants displayed the highest expression level compared to dark- and FML-treated plants.

### 2.4. FML Enhances Growth of Indian Mustard Seedlings

To assess the biological significance of FML on plants, we selected the Indian mustard (*B. juncea*) crop plant for this analysis because of its fast, high and homogenous seed germination. Accordingly, 10-day-old mustard seedlings were exposed to FML for three consecutive nights (starting a day before the FM; 5 h each night), and their growth parameters were recorded after 1 and/or 2 weeks and compared to dark-treated plants. All experiments (three experiments, each having four repeats) displayed similar results, and a representative one is shown in Figure 4. Seedling performance after 2 weeks was significantly improved under FML compared to the dark (Figure 4A). Seedling fresh and dry weights were significantly higher under FML than dark treatment (Figure 4B,C), and similarly, the fresh and dry weights of the roots were significantly higher in seedlings exposed to FML than in seedlings subjected to the dark (Figure 4D,E). This suggests that FML has a positive effect on the post-germination growth of Indian mustard seedlings.

## 3. Discussion

Many creatures on Earth can sense the solar radiation reflected by the moon. Moonlight and the lunar cycle can affect the behavior of vertebrate and invertebrate species including reproduction, communication, foraging and predation [2]. Rhythmic exposure to moonlight is believed to affect the life cycle of plants, from seed germination to fruit maturation. Notably, lunar farming is still practiced in certain places around the world where farmers are using the lunar cycle to organize their agricultural tasks. A comprehensive examination of the literature linking agricultural practices and lunar phases from a scientific viewpoint has led to the conclusion that in most cases, lunar farming has no scientific support [5]. Yet, the results presented here show that plants do respond to FML and significantly change their nuclear morphology and their proteomic and metabolomic profiles, which might profoundly affect plant performance. Our results are consistent with a previous report addressing the effect of moonlight on gene expression in *C. arabica* and confirming the notion put forward by Breitler et al. [14] that the moonlight is perceived as a stress signal. Furthermore, exposure of mustard seedlings to FML for 3 consecutive nights significantly enhanced all growth parameters examined, thus providing scientific support for lunar farming. Our results are consistent with early studies demonstrating the positive impact of moonlight on germination, plant growth and harvest [7,8]. Thus, as noted previously, various aspects of plant growth and development (e.g., germination, vegetative growth, flowering) may be significantly influenced by the state of the moon at the time of sowing [6].

### 3.1. FML Is Perceived as a Signal Promoting Activation of Stress-Associated Substances

The variation in proteome and metabolome profiles induced by FML appear to be characteristic of plants responding to a variety of stress conditions. The increase in free amino acids including proline, glutamine and glutamic acid and the sugar raffinose has been reported extensively as a notable response of plants subjected to multiple biotic and abiotic stresses [18,19], and the accumulation of free amino acids has been implicated in increasing tolerance to adverse environmental conditions, such as drought, heat and salinity [20]. Amino acids can act in multiple ways to mitigate stress. Thus, apart from their fundamental role as the building blocks for protein synthesis, amino acids also serve as precursors for the synthesis of secondary metabolites such as polyamines (e.g., putrescine) from arginine [21], ethylene from methionine [22] and salicylic acid via the phenylalanine pathway [23]; these secondary metabolites have been implicated in stress response and tolerance [23,24,25]. The accumulation of the most-studied amino acid proline in response to abiotic stresses has been positively correlated with plant tolerance, where proline can act as osmolytes, scavengers of ROS or molecular chaperones [26]. Thus, from the metabolic viewpoint, moonlight is perceived by plants as a “stress signal”. Interestingly, the response to moonlight has two distinguishable stages: an early “adjustment” stage occurring within 1 h of exposure to FML, which is characterized by a significant reduction in free amino acids, and a late, more “adaptable” stage characterized by a significant increase in the levels of multiple amino acids and the stress-induced sugar raffinose [19].

Proteome analysis uncovered multiple proteins whose levels were increased significantly under FML, including the photoreceptors PhyB and Phot2 and multiple stress-related proteins HSPs, chaperonins and ROS-detoxifying enzymes. In accordance with the increased levels of multiple amino acids, the proteome data listed one major enzyme involved in amino acid metabolism, which was upregulated under FML. Thus, the increased level of the amino acid glutamine (6.14-fold) is highly correlated with an increased level of the enzyme involved in its biosynthetic pathway, glutamine synthetase (8.07-fold) and the increase in its precursor, glutamate (7.44-fold). Notably, glutamine is a key amino donor for the synthesis of amino acids, nucleotides, and multiple nitrogen-containing compounds in all organisms and thus central in maintaining cellular integrity. It may also function as a signaling molecule controlling the expression of genes in plants involved in stress response [27].

### 3.2. Upregulation of Photoreceptor by FML

Of particular interest was the upregulation of photoreceptors PhyB and Phot2 in tobacco under FML. These findings are consistent with the upregulation of *PHOT1* gene in *C. arabica* as a response to moonlight [14]. PhyB is a red/far-red light-absorbing photoreceptor playing an essential role in a variety of photomorphogenic processes, such as germination, de-etiolation and flowering [28,29], as well as in plant adaptation to biotic and abiotic stress conditions [30,31]. PhyB, like other Phy photoreceptors, undergoes dynamic photoconversion between the red-light (R)-absorbing Pr and the far-red (FR)-light-absorbing Pfr forms, whereby the Pfr form is the active form [32]. The photoconversion from Pr to Pfr may lead to translocation from the cytoplasm to the nucleus, where Phys are often found in structures called photobodies [33]. The diverse light-signaling events might be driven by Pfr interactions with downstream factors, such as the PHYTOCHROME-INTERACTING FACTORs (PIFs), which function as negative regulators of light responses [29]. Dark conversion and degradation of photoactivated PhyB may be controlled by phosphorylation, which facilitates the conversion from active Pfr to the inactive Pr form, and by degradation by the proteasome system mediated by PIF5 [34]. Thus, we assume that the absence/low level of PhyB in dark-treated plants probably resulted from its dark conversion to the Pr form, which induced its PIF5-dependent degradation. The increased level of PhyB and Phot2 proteins in FML-treated plants could be explained by the transcriptional activation of both genes. Indeed, in *Arabidopsis* seedlings, *PHOT2* expression was enhanced by light [35], and our data show that the levels of *PHYB* and *PHOT2* mRNAs were increased under FML and to a higher extent under the growth room’s light. It is also possible that dephosphorylation by a phytochrome-associated serine/threonine phosphatase 2 (PAPP2/FYPP3) could increase the stability of the PhyB Pfr form [36,37] and its persistence under FML.

Phototropins are photoactivated serine/threonine protein kinases that undergo autophosphorylation at multiple sites; phosphorylation of serine residues in the kinase activation loop is most crucial for blue-light responses such as phototropism, chloroplast accumulation and stomatal opening in *Arabidopsis thaliana* [38,39,40,41,42]. As kinases, photoreceptors may exert their effect on multiple light-regulated processes by phosphorylating multiple target proteins, including PIFs and other substrate proteins that are awaiting discovery. Finally, upregulation of Phot2 under the low PAR of FML may be linked to enhancement of growth because phototropins were found to promote plant growth in response to blue light under a low-light environment [43].

### 3.3. Epigenetic Variation Induced by FML

Exposure to FML as well as to light induces, irrespective of light intensity, significant epigenetic variation, including a more than 2-fold increase in nuclear size accompanied by a reduction in CG methylation at centromeric repeats and the occurrence of the fast-migrating isoform of histone H3. This increase in volume might reflect the decondensation of chromatin, which otherwise assumes a compact configuration during the dark period. This is similar to chromatin decondensation, which occurs following the exposure of plants to various stresses, such as low light, heat, dark and protoplasting [44,45,46,47,48,49].

Our results are consistent with a previous report demonstrating changes in the nuclear architecture of *Arabidopsis* upon the transition from dark to light, which reflects genome reorganization accompanied by chromatin decondensation, possibly leading to transcriptional reprogramming associated with the establishment of photosynthesis [15]. Furthermore, analysis of dimethyl histone H3K4/K9 (K4/K9me2) revealed a fast-migrating H3K4/K9me2 isoform lacking a portion of the C-terminal region, which is present under light and disappears during the dark period but reoccurs under FML. This suggests that the light quality rather than intensity is sufficient to induce variation in DNA methylation and cleavage of histone H3 and consequently chromatin organization, which provides the chromatin environment necessary for the activation of genes associated with both photosynthesis and plant response to stress. The biological significance of C-terminus cleavage of the modified histone H3 for chromatin structure and function is presently unknown. Based on the molecular mass of the H3 fast-migrating isoforms, we assume cleavage of about 15 to 25 amino acids from H3 C terminus. Using the Swiss modeling platform [50], we showed (Appendix A) that removal of 10 or 20 amino acids of the tobacco H3.3 C terminus resulted in the loss of the α3 helix of the histone fold. This may lead to eviction from nucleosomes and replacement by a newly synthesized H3, since a histone H3 lacking the α3 helix cannot be assembled into nucleosomes [51].

The effect of light on plant chromatin organization appears to be mediated by photoreceptors and dependent on light intensity [52]. Accordingly, low light intensity triggers the decompaction of chromocenters in the rosette mesophyll cells of *Arabidopsis thaliana*, and high light often controls chromatin compaction [53,54,55]. While the blue-light receptor Cryptochrome 2 (Cry2) regulates chromatin decompaction under low light, PhyB appears to control chromatin compaction under high light intensity [54,55]. Thus, photoreceptors upregulated under FML transmit signals for epigenetic-variation-induced chromatin reorganization and gene expression, which are mediated by epigenetic modifiers.

### 3.4. Conclusions

The data presented here shed light on the potential role of moonlight, an overlooked environmental factor, in plant growth and development. Apparently, moonlight intensity, even at its peak, is insufficient for driving photosynthesis, yet it is perceived as a signal promoting extensive variation in nuclear structure and in protein and metabolite profiles, which could affect plant growth and development and response to stress. Commonly, abiotic stresses result from a deficiency or excess in environmental factors such as water and temperature and can substantially reduce plant growth and development, reproduction and survival. However, light emitted by the moon induces a stress-like response under seemingly favorable growth conditions and the absence of noticeable stress. The consequences of “stress response without stress”, that is, increased levels of free amino acids and the sugar raffinose as well as photoreceptors and stress proteins, could positively affect plant growth and development, confer stress priming [30,56,57,58,59] and may underlie lunar farming.

## 4. Materials and Methods

### 4.1. Plant Growth Conditions and Exposure to Moonlight

*Nicotiana tabacum* seeds were sown in 1 L pots containing standard gardening soil composed of peat and perlite (2:1) and grown in a growth room under a light intensity of 980 lux, 65–70% humidity, 25 +/−1 °C temperature, and 14/10 h (day/night) photoperiod conditions. Tobacco seedlings (6-week-old) were placed in the experimental site (the roof of the building was covered with asphalt sheets) 2 h after entry into the dark period and exposed for 1 and 5 h to the dark or to full moonlight (FML). Data on the FML spectrum and intensity are given in Breitler et al. [14]. The temperature (12–14 °C; during the night of 17 March 2022) at the experimental location was monitored using a USB iButton Reader, DS9490# (MAXIM, China). Leaf samples were collected after 1 and 5 h of exposure to FML or to darkness (4 biological replicates), frozen immediately in liquid nitrogen or fixed in acetic acid–ethanol (1:3 *v/v*) and kept at −80 °C or −20 °C, respectively, until further analysis.

We selected a crop plant, *Brassica juncea* (Indian mustard), for assessment of the effect of FML on post-germination growth. Seeds were sown in standard gardening soil in small pots under the conditions described above. Seedlings (10-day-old) were subjected to FML or darkness for three consecutive nights (5 h each night) starting a day before the full moon, followed by 1 or 2 weeks of growth in the growth room. Plants were harvested and analyzed for fresh and dry weights of the whole plant and the root system. The experiment was performed three times, each with 4 repeats, during 13–15 June, 11–13 August and 10–12 September 2022.

In a complementary experiment, tobacco seedlings were subjected to the new moon (dark night; 26 September 2022) to assess for potential light pollution at the experimental site, followed by analysis of nuclear size and primary metabolites.

### 4.2. Nuclei Isolation and Confocal Microscope Inspection

Nuclei were prepared from leaves of tobacco or *B. juncea* using the method described by Saxena et al. [60]. Briefly, leaves were chopped using a razor blade in a nuclei isolation buffer (NIB) (10 mM MES-KOH, pH 5.5, 0.2 M sucrose, 2.5 mM EDTA, 2.5 mM dithiothreitol, 0.1 mM spermine, 10 mM NaCl, 10 mM KCl, 0.15% Tritron X-100). The homogenate was gently stirred for 45 min at 4 °C and filtered through 100 μm nylon mesh followed by 30 μm nylon mesh. The filtered extract was centrifuged for 8 min at 2000 RPM at 4 °C. The pellet was gently washed to remove the upper chloroplast layer, and nuclei pellets were recovered and washed twice with NIB buffer, fixed in ethanol–acetic acid (3:1 *v/v*) and stored at −20 °C until further use. Nuclei were stained for 10 min with 10 μg/mL diamidino-phenyl-indole (DAPI), washed twice with 2× SSC and mounted in Vectashield (Vector Laboratories, Burlingame, CA, USA). Nuclei size measurements were carried out using a confocal microscope (Zeiss LSM 900), and the data were processed using Excel software (Microsoft, Redmond, WA, USA).

### 4.3. Acid Extraction of Proteins and Immunoblotting

Leaf samples of tobacco plants exposed to 1 and 5 h of darkness, FML and light were extracted with 3% trichloroacetic acid (TCA) in NETN buffer (100 mM NaCl, 1 mM EDTA, 20 mM Tris, pH 8, and 0.5% NP-40) supplemented with a protease inhibitor cocktail (Sigma, St. Louis, MO, USA). Protein concentration was determined via the Bradford reagent. Acid-soluble proteins (2.5 µg) enriched with histones were resolved by 15% SDS/PAGE gel and immunoblotted with antibodies (Cell Signaling Technology, Danvers, MA, USA), namely, anti-dimethylated H3K4 (#9725), anti-dimethylated H3K9 (#9753) and anti-H3-C-terminal (D1H2, #4499). The membrane was washed 3 times with TBST for 5 min followed by incubation with secondary antibody of goat anti-rabbit HRP conjugate for 1 h. The membrane was washed 3 times with TBST for 5 min and incubated in Super Signal West Pico Chemiluminescent Substrate (Thermo Fisher Scientific) for 1 min to visualize and image the protein bands in the chemiluminescence imager (Chemi-DoC, Bio-rad).

### 4.4. DNA Extraction and Methylation Analysis

For DNA methylation analysis, genomic DNA was extracted from leaves of tobacco, as reported previously [61]. DNA was further treated with RNase A to remove RNAs followed by chloroform-isoamyl alcohol extraction and ethanol precipitation. DNA was dissolved in 100 μL of H_2_O and protein contaminants were removed via the salting-out procedure by adding 50 μL of 7.5 M and ammonium acetate and carrying out incubation on ice for 20 min. Samples were centrifuged at high speed for 10 min, the supernatant was collected and DNA was precipitated by adding 2 volumes of 100% cold ethanol followed by centrifugation at high speed for 15 min. DNA was dissolved in H_2_O and quantified by measuring absorbance at 260 nm using a nanodrop ND-1000 spectrophotometer. The quality of DNA was checked using ethidium bromide (EtBr) staining after running on 1.0% agarose gel.

DNA methylation analysis was performed using Chop PCR (methylation-sensitive enzyme digestion followed by PCR). To this end, genomic DNA (1 μg) was digested with the methylation-sensitive enzymes *Hpa*II and *Msp*I, followed by PCR to amplify various DNA sequences. We used the following primers: HRS60-F, GATCCATCCGGGCCCAAGGCGG; HRS60-R, CGTCGTGGAATCGCCTAATATTTG; RS-3-19-F, CATCTCTGTATAAACGATCCGATCG; RS-3-19-R, CAACAATTTGAATCCCATGAAATCG; Tto1-F, CGCTGTGCAGTAGTGTTTAGTGC, and Tto1-R, CAGGTTTCTGAGAACTGAACAC. The PCR amplifications were performed in 20 μL of reaction mixture containing 10 μL of 2× Taq PCR Master Mix (TIANGEN, Beijing, China), 500 nM primers and 50 ng template DNA. The amplification was performed in a Bio-rad T100 thermal cycler using the following program: 95 °C for 3 min, 35 cycles at 95 °C for 30 s, 57 °C for 30 s and 72 °C for 2 min, which was followed by a final extension at 72 °C for 5 min. The PCR products were resolved on a 1.0% agarose gel and visualized via EtBr staining.

### 4.5. Metabolite Analysis

Gas chromatography–mass spectrometry (GC-MS) was used to quantify primary metabolites in six separate replicates, as described previously [62]. Briefly, lyophilized leaf samples were extracted with a precooled mix containing methanol, chloroform, and MiliQ H_2_O (2.5:1:1 *v/v*/*v*, respectively) supplemented with sorbitol as the internal standard (4.5 µg/mL), vortexed and incubated for 10 min at 25 °C on an orbital shaker. Samples were sonicated for 10 min in an ultra-sonication bath at room temperature followed by centrifugation at high speed (10 min, 16,000× *g*). The supernatant was collected, with the addition of 300 µL of MiliQ H_2_O and 300 µL of chloroform, vortexed for 10 s and centrifuged for 5 min at high speed. The upper phase was collected, lyophilized and subjected to derivatization. Derivatization was performed by adding 40 µL of methoxyaminhydrochloride (20 mg/mL in pyridine) to the lyophilized samples followed by incubation for 2 h at 37 °C on an orbital shaker. Then, 70 µL of MSTFA and 7 µL of an alkane mix were added and incubated for 30 min at 37 °C with constant shaking. Samples were subjected to gas chromatography–mass spectrometry (GC-MS) analysis (Agilent Ltd., Santa Clara, CA, USA), as described previously [62,63]. Separation was carried out on a Thermo Scientific DSQ II GC/MS in electron ionization (EI) mode, using a Factor Four capillary VF-5 ms column (Agilent Ltd., Santa Clara, CA, USA). The acquired chromatograms and mass spectra were evaluated using Xcalibur (version 2.0.7) software, and metabolites were identified and annotated using the Mass Spectral and Retention Time Index libraries available from the Max-Planck Institute for Plant Physiology, Golm, Germany (http://csbdb.mpimp-golm.mpg.de/csbdb/gmd/msri/gmd_msri.html, accessed on 11 May 2021). A metabolite was considered differentially present if it had an unadjusted *p* value < 0.05 and fold change > 1.3. The metabolite level was calculated by normalizing the intensity of the peak of each metabolite to the sorbitol standard. PCA, ANOVA, Student’s *t*-tests and hierarchical clustering were performed using Metaboanalyst 4.0 [64].

### 4.6. Proteome Analysis

For proteomic analysis, 3 replicates of 10 mg of ground leaves derived from plants treated with 5 h of darkness and 5 h of FML were placed in 2 mL tubes and incubated with 100 µL of NETN buffer (100 mM NaCl, 1 mM EDTA, 20 mM Tris, pH 8.0 and 0.5% NP-40) at 4 °C for 6 h with gentle rotation, and then centrifuged at 4 °C at high speed for 10 min. Altogether, 50 µL of supernatants was collected and stored at −20 °C until its use for comparative, quantitative proteome analysis. Proteome analysis was performed using the proteomic services of the Smoler Protein Research Center at the Technion, Haifa, Israel, using LC-MS/MS on LTQ Orbitrap (ThermoFisher Scientific, Waltham, MA, USA; https://proteomics.net.technion.ac.il/proteomic-services/, accessed on 11 May 2021). Protein samples were first subjected to a depletion procedure for the major highly abundant protein in plant leaves, namely, Ribulose-1,5-bisphosphate carboxylase/oxygenase (RuBisCO), to avoid hindering low-abundance proteins. Protein identification and quantification were carried out with MaxQuant, using *Nicotiana tabacum* proteins from uniport as a reference. Quantification and normalization were performed using the LFQ method. Subsequent bioinformatics analysis was performed using Perseus software [65]. Proteins marked as “contaminant “and “only identified by site” were filtered out. In an additional step, only proteins in which at least one of the groups had at least 2 non-zero replicates, proteins having at least 2 peptides and a peptide sequence coverage of more than 20% were retained. A protein was considered differentially expressed if it had a nominal *p* value < 0.05 and absolute fold change > 1.3. PCA and Student’s *t*-test analysis were performed using Metaboanalyst 4.0 [64].

### 4.7. RNA Analysis

RNA levels of tobacco *PHYB* (Gene ID, LOC107767359) and *PHOT2* (Gene ID, LOC107793292) were analyzed via semiquantitative PCR followed by real-time PCR. To this end, total RNAs were extracted from tobacco leaves subjected to the dark, FML or growth room light using Bio-Tri reagent (Bio-labs, Israel) using the manufacturer’s protocol. RNA was treated at 37 °C for 15 min with RNase-free DNase (Promega) followed by phenol/chloroform extraction and ethanol precipitation. The RNA samples were quantified with a NanoDROP ND-1000 spectrophotometer (Thermo Fisher Scientific). The total RNA (1 μg) was reverse-transcribed with the qPCRBIO cDNA Synthesis Kit (PCR Biosystems, USA), according to the manufacturer’s instructions. The resulting cDNA was used as the template for semiquantitative PCR (Semi-qPCR) and quantitative PCR (qPCR). The cDNA was used as a template for semiquantitative PCR by using PCR master mix (Hylabs), and resultant fragments were resolved in 3% agarose gel containing ethidium bromide. The qPCR analysis was carried out in the Bio-Rad CFX96^TM^ Real-Time PCR Detection System, with powerSYBER Green PCR Master Mix (Thermo Fisher Scientific). The reaction mixture consisted of 5 μL of SYBR Green Master Mix, 0.5 μL (250 nM) of each of the forward and reverse gene-specific primers, and 4 μL of diluted template cDNA. The PCR was performed under the following conditions: initial denaturation at 95 °C for 3 min, followed by 39 cycles of 95 °C for 10 s, 60 °C for 30 s and 72 °C for 20 s. Three biological replicates per reaction and no template control were included in every PCR run. The amplification efficiency of the primer pairs was evaluated by generating standard curves of the linear amplification of the target genes from reactions containing serial dilutions of the cDNA template. A melting curve analysis was performed to check primer specificity and the amplification of a single product. The expression levels of the target genes were normalized to the expression of a known housekeeping gene encoding elongation factor 1α. We designed primers to distinguish between a PCR product derived from DNA and one derived from cDNA. All primers were first approved for their ability to direct PCR using genomic DNA as a template. The primers used for gene expression analysis were qPhyB-F-TGGTACGGTATGCACCATCAC, qPhyB-R-GCAAGCAGTGGCTTCTTTAC, qPhot2-F-GCAAGCAGTGGCTTCTTTAC, qPhot2-R-TCTTTCTGGTCTGTATCTTTCCC, qEF-1α-F-ACGCACTGCTTGCTTTCA and qEF-1α-R- AACCTCCTTCACGATTTCAT. The statistical analysis was performed using the 2^−ΔΔCT^ method [66].

## Figures and Tables

**Figure 1 plants-12-01121-f001:**
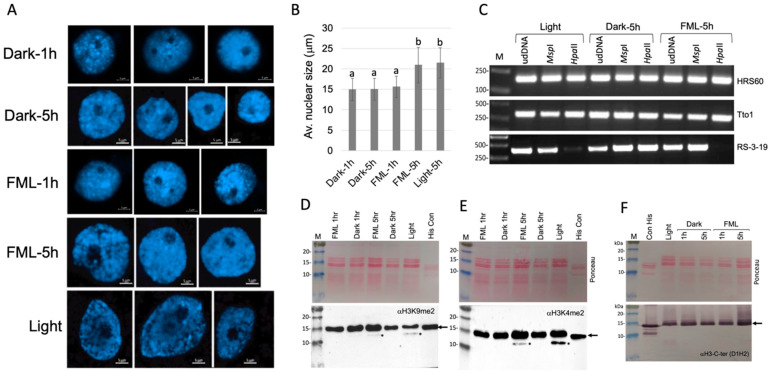
FML induces changes in nuclear size. (**A**) Leaves of tobacco plants exposed to dark, light or FML for 1 h or 5 h were fixed in acetic acid–ethanol (1:3), and nuclei were prepared, stained with DAPI and visualized under a confocal microscope. (**B**) Average diameter of nuclei prepared from dark-, light- and FML-treated plants (*n* = 100). Vertical bars represent the standard deviation. Statistical significance was determined with a One-Way ANOVA Calculator, Including Tukey HSD (Social Science Statistics). Different letters indicate statistically significant differences between treatments (*p* < 0.01). (**C**) Exposure to light and FML was accompanied by a reduction in CG methylation at centromeric repeats. Genomic DNA extracted from plants exposed to light, 5 h of dark and 5 h of FML was digested with *Hpa*II or *Msp*I and subjected to PCR to amplify HRS60, Tto1 and RS-3-19 sequences. Undigested DNA (Ud DNA) was used as a reference. The experiment was repeated 3 times. M indicates the DNA size marker given in base pairs. (**D**–**F**) Light and FML induced C-terminal cleavage of histone H3. Acid-soluble fractions from leaves derived from light-, dark- and FML-exposed tobacco plants were analyzed via immunoblotting to detect the indicated modified histone H3, dimethyl H3K9 (αH3K9me2) (**D**), dimethyl H3K4 (αH3K4me2) (**E**) and histone H3 using antibody to the C-terminal region (αH3-C-ter) (**F**). Upper panels show Ponceau staining of membranes. Asterisks indicate the fast-migrating isoforms and arrows the major H3 isoforms. Con His represents control histone proteins from calf thymus.

**Figure 2 plants-12-01121-f002:**
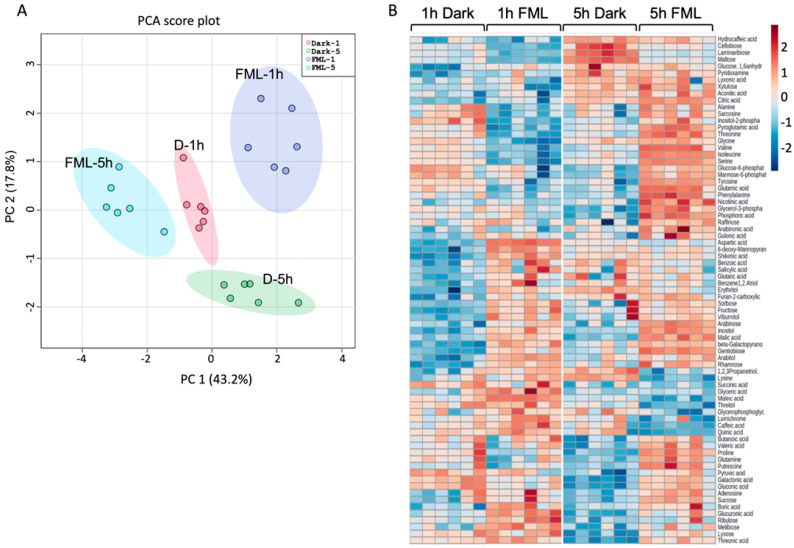
FML induces changes in primary metabolite profile of tobacco plants. (**A**) PCA score plot comparing metabolite profiles of tobacco leaves after exposure for 1 h and 5 h to darkness or FML. (**B**). Hierarchical clustering of differentially expressed metabolites. Each treatment is represented by six columns (six repeats).

**Figure 3 plants-12-01121-f003:**
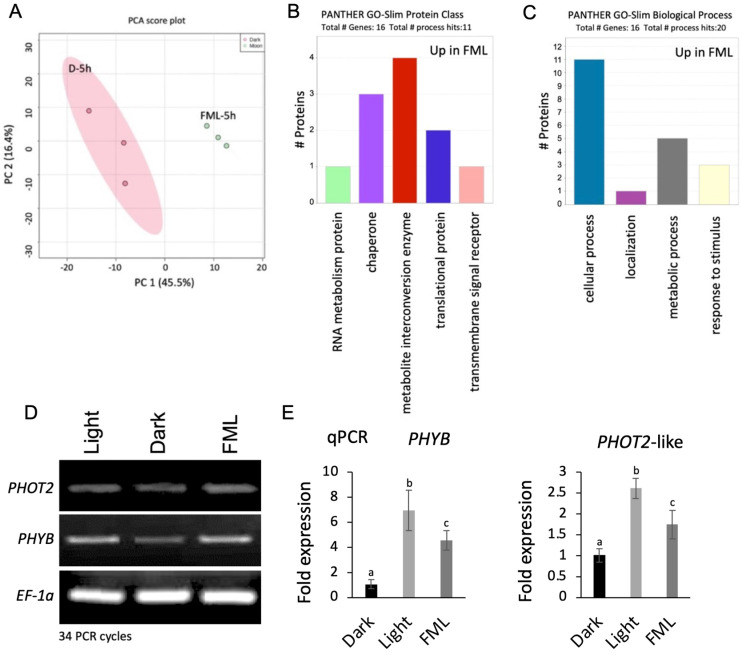
Proteome analysis of leaves of tobacco plants exposed to darkness and FML. (**A**) Principal component analysis (PCA) score plot comparing the proteome profiles between leaves derived from plants subjected to 5 h of darkness and 5 h of FML. (**B**) Categorization analysis (protein class) of the 17 DEPs upregulated under FML treatment. (**C**) Biological process. Categorization analysis was performed using a PANTHER bioinformatic. (**D**) Semiquantitative PCR demonstrating the expression of *PHYB* and *PHOT2* genes under light, darkness and FML. (**E**) Real-time PCR (qPCR) showing the fold expression of *PHYB* and *PHOT2* under dark, light and FML. EF1α was used as a control gene. Statistical significance was performed using a One-Way ANOVA Calculator, Including Tukey HSD (Social Science Statistics). Different letters indicate statistically significant differences between treatments (*p* < 0.05). qPCR was performed twice, each with 3 replicates.

**Figure 4 plants-12-01121-f004:**
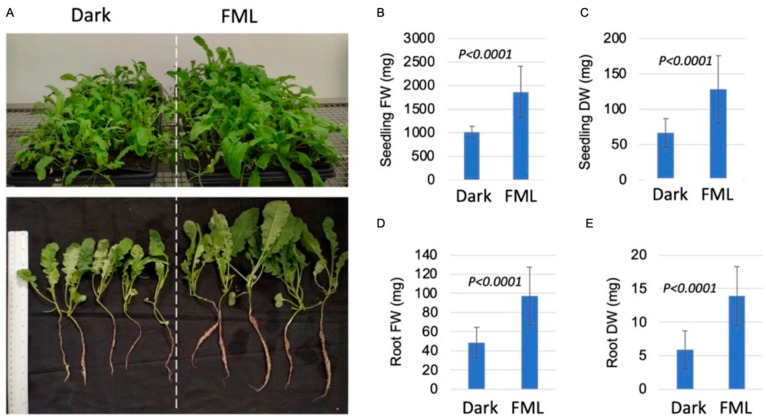
Exposure of *B. juncea* seedlings to FML enhances their growth. Ten-day-old seedlings were subjected for 3 consecutive nights to 5 h of FML (starting a day before the FM), after which seedlings were placed in a growth room, and their growth was recorded after 2 weeks. (**A**) The appearance of FML- and dark-treated seedlings, 2 weeks after treatment. (**B**) Average seedling fresh weight (FW). (**C**) Average seedling dry weight (DW). (**D**) Average root FW. (**E**) Average root DW. Vertical bars represent the standard deviation (*n* = 50). *p* value was determined using Student’s unpaired *t*-test using GraphPad software.

**Table 1 plants-12-01121-t001:** A list of differentially expressed metabolites in tobacco leaves after 1 h and 5 h of exposure to FML and the dark. FC, fold change > 1.3 (*p* value < 0.05).

1 h Exposure to FML vs. Dark			5 h Exposure to FML vs. Dark		
Metabolite	FC (FML/D)	Up/Down	Metabolite	FC (FML/D)	Up/Down
Amino Acids			Amino Acids		
Alanine	0.46	Down	Alanine	1.70	Up
Aspartic acid	2.49	Up	Glutamic acid	7.44	Up
Glutamic acid	0.32	Down	Glutamine	6.14	Up
Glutamine	0.58	Down	Isoleucine	2.11	Up
Glycine	0.32	Down	Lysine	0.70	Down
Isoleucine	0.50	Down	Phenylalanine	3.54	Up
Phenylalanine	0.70	Down	Proline	3.93	Up
Proline	0.56	Down	Serine	2.04	Up
Serine	0.41	Down	Threonine	1.80	Up
Threonine	0.62	Down	Tyrosine	5.58	Up
Tyrosine	0.43	Down	Valine	2.03	Up
Valine	0.61	Down			
Sugars			Sugars		
6-deoxy-Mannopyranose	1.41	Up	Cellobiose	0.73	Down
β-Galactopyranosyl-1,3-arab. abinose	1.62	Up	Gentiobiose	1.41	Up
Fructose	1.32	Up	Glucose-6-phosphate	1.34	Up
Gentiobiose	1.39	Up	Glycerol-3-phosphate	1.37	Up
Inositol-2-phosphate	0.75	Down	Laminaribiose	0.61	Down
Ribulose	1.50	Up	Maltose	0.67	Down
Sorbose	1.31	Up	Mannose-6-phosphate	1.42	Up
Viburnitol	1.31	Up	Raffinose	2.09	Up
Other metabolites			Other metabolites		
Boric acid	1.64	Up	Boric acid	1.40	Up
Caffeic acid	1.45	Up	Butyric acid	2.04	Up
Quinic acid	1.48	Up	Phosphoric acid	1.39	Up
Sarcosine	0.48	Down	Putrescine	1.52	Up
Shikimic acid	1.33	Up	Quinic acid	0.25	Down
			Sarcosine	1.70	Up

**Table 2 plants-12-01121-t002:** A partial list of proteins up- and downregulated under FML compared to the dark. FC, fold change (>2; *p* < 0.05). Chl, chloroplast; CW, cell wall; Cyt, cytoplasm; Mit, mitochondria; Mem, membrane; nd, not defined; Nuc, nucleus.

Gene Names	Protein Names	Avg (FML)	Avg (Dark)	FC (FML/Dark)	Adj *p* Value	Cell Comp.
LOC107767359	Phytochrome B	9.0 × 10^8^	2.9 × 10^7^	30.627	0.00614	Nuc; Mem
LOC107777858	Glutamine synthetase	9.1 × 10^8^	1.1 × 10^8^	8.152	0.00618	Cyt.
LOC107775921	Heat shock 70 kDa protein	8.3 × 10^9^	1.4 × 10^9^	5.927	0.00535	Mem; Mit
LOC107828946	Peroxidase	7.1 × 10^8^	1.3 × 10^8^	5.664	0.00906	CW
LOC107790714	Heat shock 70 kDa protein 15-like	8.5 × 10^9^	1.6 × 10^9^	5.298	0.00535	Nuc; Mem; Cyt
LOC107769513	PSI subunit V	7.0 × 10^9^	1.5 × 10^9^	4.719	0.00822	Chl
LOC107781119	LHCP translocation defect-like	5.9 × 10^9^	1.3 × 10^9^	4.467	0.00454	Chl
LOC107769068	Heat shock protein 90	9.2 × 10^8^	2.1 × 10^8^	4.418	0.04268	nd
LOC107763119	23 kDa subunit of OES of photosystem II	7.0 × 10^9^	1.8 × 10^9^	3.818	0.03765	Chl
LOC107770030	60S ribosomal protein L30	1.1 × 10^8^	3.0 × 10^7^	3.558	0.03079	Cyt
LOC107793292	Phototropin-2	7.7 × 10^8^	2.2 × 10^8^	3.531	0.00657	Nuc; Cyt
LOC107810219	Chaperonin 60 subunit beta 2	7.7 × 10^10^	2.5 × 10^10^	3.046	0.00736	Chl
LOC107806960	Superoxide dismutase	8.1 × 10^9^	2.9 × 10^9^	2.784	0.00535	Chl
LOC107778847	Malic enzyme	7.8 × 10^9^	2.9 × 10^9^	2.657	0.00951	Mem; Chl
LOC107820445	Chaperone protein dnaJ A6	6.0 × 10^8^	2.3 × 10^8^	2.596	0.02283	Cyt
LOC107825601	Aconitate hydratase	9.9 × 10^9^	4.1 × 10^9^	2.425	0.00005	Cyt
LOC107768788	60S ribosomal protein L3-2-like	1.1 × 10^8^	5.1 × 10^7^	2.196	0.02283	Cyt
LOC107785603	F-box protein PP2-B11-like	2.5 × 10^8^	5.2 × 10^8^	−2.08	0.00454	nd

## Data Availability

The data supporting the findings of this study are presented in the main text and in the supporting information of this article.

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
