# Peer review of "Moonlight Is Perceived as a Signal Promoting Genome Reorganization, Changes in Protein and Metabolite Profiles and Plant Growth"

_plants, 2023, doi:10.3390/plants12051121_

Round 1
Reviewer 1 Report
It is an interesting study on the effect of moonlight on plant growth at the cellular, transcriptional and metabolic-omic levels. I was suspicious about the whole idea of the study and the validity of the experimental design at first. Yet, after reading the ms., I did not find any significant pitfall in the experimental design though.
One suggestion is to delete the “stress” word in the title. Based on all the presented data, I am not convinced that those alterations necessarily mean 'stress' to the plant.
Author Response
Reviewer 1:
It is an interesting study on the effect of moonlight on plant growth at the cellular, transcriptional and metabolic-omic levels. I was suspicious about the whole idea of the study and the validity of the experimental design at first. Yet, after reading the ms., I did not find any significant pitfall in the experimental design though.
Comment: One suggestion is to delete the “stress” word in the title. Based on all the presented data, I am not convinced that those alterations necessarily mean 'stress' to the plant.
Response: We omitted the ‘stress’ word from the title. We do agree that those alterations do not necessarily mean 'stress' to the plant, they just illustrate the mode of response often seen when plants encounter stress.
Reviewer 2 Report
I don't think this study have any scientific significance.
Author Response
Reviewer 2:
Comment: I don't think this study have any scientific significance.
Response: No response!
Reviewer 3 Report
The manuscript ‘The moonlight is perceived as a stress signal inducing genome reorganization, changes in protein and metabolite profiles and 3 plant growth’ by Singiri et al conducted a biochemical and morphological characterization of the effect of full moonlight (FML) in plant cell biology and development.
The experimental design, data analyses, and result presentation, are basically reasonable, though I think some information must be added to the manuscript:
1- I could not find any information about how the control experiments (dark and light) were performed. I am specially concern about how environmental parameters other that light (such as temperature) were homogenized in the whole set of experiments.
2- I would like to have more details about how the metabolomics analysis was performed. What kind of ionization source was used (EI or APCI)? Were all the identified metabolites finally annotated or some of them remain unidentified? I also wonder why not to analyze the data directly using a multivariate approach (PCA or PLS-DA), instead of using first a univariate approach (p-value) and then a multivariate analysis (PCA)?
3- Please clarify the paragraph in line 111. Specially in the sentence ‘Interestingly, during the dark period 115 H3FMI disappeared, but re-established upon exposure to FML for 5h but not for 1h’ I am confused to which protein authors refer to.
Minor changes:
- Table 1. Please reduce the number of decimals to significant figures.
- All along the paper, please use the symbol ° to indicate Celsius degrees instead of º
- Line 419, change ‘hr’ for ‘h’
- Line 77, change ‘Brassiplant ca juncea’ for ‘Brassica juncea’
Author Response
Reviewer 3:
Comment 1:
I could not find any information about how the control experiments (dark and light) were performed. I am specially concern about how environmental parameters other that light (such as temperature) were homogenized in the whole set of experiments.
Response to comment 1:
These experiments were performed under natural moon light (on the roof of the building as described in M&M) where control plants were placed in the dark next to the FML-exposed plants (The temperature of both were monitored continuously). The light grown plants (growth room under 25oC) were used as a reference (Conditions are given in M&M).
Comment 2-1: I would like to have more details about how the metabolomics analysis was performed. What kind of ionization source was used (EI or APCI)?
Respone to comment 2-1: Ionization was performed by electron ionization (EI), which is now indicated in M&M.
Comment 2-2: Were all the identified metabolites finally annotated or some of them remain unidentified?
Response to comment 2-2: Being untargeted metabolic analysis, not all metabolic signals could be annotated. Actually, among the large number of signals that potentially associated with metabolites, only 98 metabolites were annotated in the present work. Multivariant analysis was performed only on annotated metabolites.
Comment 2-3: I also wonder why not to analyze the data directly using a multivariate approach (PCA or PLS-DA), instead of using first a univariate approach (p-value) and then a multivariate analysis (PCA)?
Response to comment 2-3: Indeed, we have displayed a multivariate approach (PCA) on the differentially expressed metabolites passing the p value<0.05. We now add multivariant analysis (PCA) for all annotated metabolites and the data was added to supplementary materials (Fig. S2).
Comment 3: Please clarify the paragraph in line 111. Specially in the sentence ‘Interestingly, during the dark period 115 H3FMI disappeared, but re-established upon exposure to FML for 5h but not for 1h’ I am confused to which protein authors refer to.
Response to comment 3: In figure 1D and 1E, immunoblotting with antibodies to dimethyl-H3K4 and dimethyl-H3K9 revealed two isoforms, a major one migrating to position of the expected protein size (marked by arrow) and a fast-migrating isoform (FMI) marked by asterisk, which is apparent under light and FML but not during the dark period. We rephrased this sentence to make it clearer.
Minor changes:
- Table 1. Please reduce the number of decimals to significant figures.
Numbers in Table 1 were reduced as suggested.
- All along the paper, please use the symbol ° to indicate Celsius degrees instead of º
Symbol for Celsius was given ° as suggested throughout the manuscript.
- Line 419, change ‘hr’ for ‘h’
The hr. was changed to h.
- Line 77, change ‘Brassiplant ca juncea’ for ‘Brassica juncea’
Changed to Brassica juncea